# Southern Sea Otter Rehabilitation: Lessons and Impacts from the Monterey Bay Aquarium

**Leilani Konrad** [1,*]**, Jessica A. Fujii** [1] **, Sandrine Hazan** [1]**, Andrew B. Johnson** [2]**, Karl A. Mayer** [1]**,
Michael J. Murray** [1] **, Teri E. Nicholson** [1]**, Michelle M. Staedler** [1] **and Colleen Young** [3]

1   Monterey Bay Aquarium, 886 Cannery Row, Monterey, CA 93940, USA
2   Defenders of Wildlife, P.O. Box 401, Folsom, CA 95763, USA
3   Marine Wildlife Veterinary Care and Research Center, Office of Spill Prevention and Response,
     California Department of Fish and Wildlife, Santa Cruz, CA 95060, USA
*   Correspondence: lkonrad@mbayaq.org

**Abstract:** As biodiversity continues to decline across the globe, conservation of wildlife species and the ecosystems they inhabit is more important than ever. When species dwindle, ecosystems that depend on them are also impacted, often leading to a decrease in the life-giving services healthy ecosystems provide to humans, wildlife, and the global environment. Methods of wildlife conservation are complex and multi-faceted, ranging from education and advocacy to, research, restoration, and rehabilitation. Here, we review a conservation program focused on helping recover the federally listed threatened southern sea otter (*Enhydra lutris nereis*) population. We describe the development of unique rehabilitation methods and steps taken to advance the program's conservation impact. Understanding this evolution can inform conservation efforts for other vulnerable species and their ecosystems.

**Keywords:** aquarium; collaboration; southern sea otter; surrogacy; wildlife rehabilitation

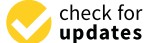



## 1. Introduction

Conserving wildlife and ecosystems is increasingly important as biodiversity continues to decline across the planet. While the intrinsic value of wildlife is clear, key species are also vital in maintaining functioning ecosystems. One potential method to help restore diminished wildlife populations and their ecosystems is the rehabilitation and release of orphaned, ill, or injured animals [1]. Despite its highly complex and contextual nature, wildlife rehabilitation can serve as a conservation tool to support the recovery of threatened populations [1,2]. Understanding how the Monterey Bay Aquarium's Sea Otter Program has evolved to enhance conservation outcomes can inform future restoration efforts for other threatened species and their ecosystems.

### 1.1. Wildlife Rehabilitation

The treatment and care of injured, diseased, and displaced indigenous wildlife, and their subsequent release as healthy animals to native habitats in the wild [3] is a broadly used conservation strategy for marine and terrestrial species. Thousands of wildlife rehabilitation programs exist globally, and the practice continues to grow [4,5]. These programs include facilities and dedicated personnel to care for ill or injured wildlife, strategies to galvanize public interest in the welfare of local wildlife populations, opportunities to advance species-specific husbandry and veterinary care and methods to identify threats to wildlife populations [1].

While most of these rehabilitation programs have a common goal of supporting wildlife, their focus extends from the rescue and release of individual animals [6,7] to larger-scale reintroduction projects of threatened and endangered species [8,9]. This broad range in program focus is often influenced by available resources and the rehabilitation

requirements of target animals. Because of their species-specific expertise, life-support systems and facilities, and financial resources, some zoos and aquariums are well suited to engage in wildlife rehabilitation. However, to achieve broader conservation goals, programs may need to establish partnerships to leverage external expertise and resources. These partnerships ultimately share or offset heavy fiscal burdens, inform strategies that minimize threats to individual animals or the wild population, and assist in identifying population-level conservation impacts.

By establishing these partnerships, a growing number of rehabilitation programs are demonstrating their effectiveness beyond individual animal welfare and measuring benefits to wild populations through post-release monitoring. For example, the California condor (*Gymnogyps californianus*) recovered from the brink of extinction through rescue, rehabilitation, and novel captive breeding and reintroduction techniques developed among zoos, non-profit organizations, and government agencies [10]. In Florida, the Manatee Rescue, Rehabilitation, and Release Program (MRP) is comprised of zoos and aquariums, agencies, academic institutions, and non-profit organizations. Through increased post-release monitoring and strategic interventions by MRP, 92% of subadult and adult West Indian manatees (*Trichechus manatus*) and a relatively high proportion of calves were rescued and released over a twenty-six-year period and successfully acclimated to the wild [11]. In the Hawaiian islands, approximately 30% of the endangered Hawaiian monk seal (*Neomanachus schauinsalndi*) population in 2012 was alive as the direct result of either opportunistic interventions or rescue and treatment by cooperating federal and state agencies and nonprofit partners [12]. These partnerships are examples of how rehabilitation programs can provide measurable benefits to wild populations.

### 1.2. The Southern Sea Otter

Following near-extirpation during the 18th and 19th century maritime fur trade, the southern sea otter (*Enhydra lutris nereis*) population has grown to 3000 individuals throughout central California [13]. Despite this increase, the current population only inhabits 13% of its historical range from Oregon to central Baja, Mexico (Figure 1) [14]. Since 1977, the southern sea otter remains listed as "threatened" under the Endangered Species Act (ESA), and the population's status and recovery plan are managed by the USFWS under the authority of the Marine Mammal Protection Act.

Sea otters are considered a keystone species, playing a significant role in restoring and maintaining the resilience of seagrass and kelp forest ecosystems through cascading trophic relationships with their invertebrate prey [15–20]. Because they reside at the interface between land and nearshore coastal waters, sea otters are susceptible to a wide variety of natural and anthropogenic threats, such as shark bites, parasites, toxins, and infectious diseases that may lead to stranding [21,22]. During their first six months, sea otter pups depend on their mothers to nurse and nourish them, and teach them survival skills, such as foraging, grooming, socializing, and avoiding threats [23,24]. If a sea otter mother cannot find enough food to meet the high energetic requirements of pup rearing [23] or becomes ill or injured, she may abandon her pup prematurely [25,26], threatening its survival without human intervention.

Determining trends in sea otter strandings (e.g., cause and demography) is critical to identifying threats to wild populations. The California Department of Fish and Wildlife (CDFW; formerly CA Department of Fish and Game) began responding to and systematically documenting southern sea otter strandings in 1968. Since then, a network of collaborators including CDFW, United States Geological Survey (USGS), Monterey Bay Aquarium (MBA), and the Marine Mammal Center (TMMC) have worked together to collect and examine stranded sea otters. Most sea otter carcasses receive a necropsy by CDFW pathologists who, if possible, determine a primary cause of death. Sea otters that strand alive are generally collected by MBA, TMMC, or CDFW and evaluated at MBA or TMMC. TMMC and MBA are the only permitted facilities that currently rehabilitate southern sea otters in California. Therefore any sea otters who are candidates for rehabilitation

are reared and cared for by TMMC and MBA until they are released back to the wild or deemed non-releasable and transferred to a long-term care facility.

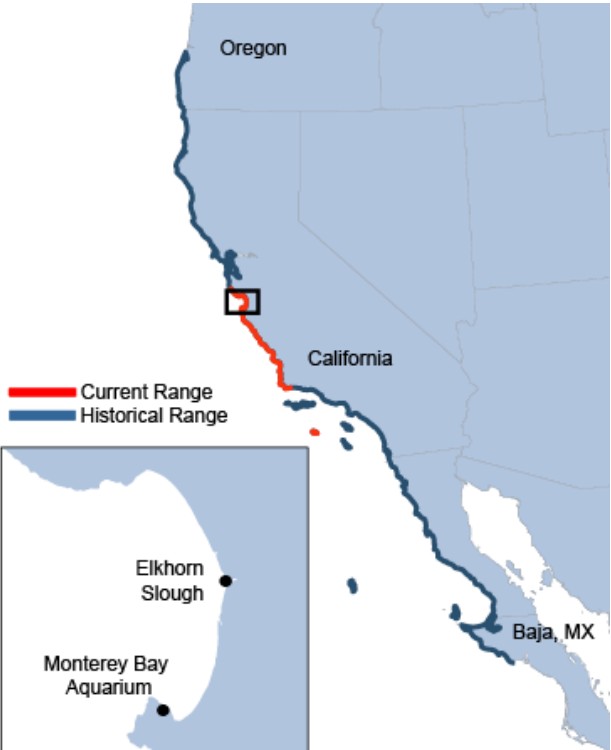

**Figure 1.** The current and historical range extent for the southern sea otter subspecies and (inset) location of Elkhorn Slough in relation to the Monterey Bay Aquarium.

For nearly four decades, the Monterey Bay Aquarium's Sea Otter Program has rescued, treated, and released stranded southern sea otters [16]. Reviewing this program's evolution highlights how wildlife rehabilitation potentially can support not only species recovery, but also ecosystem restoration more broadly and affirms how partnerships and collaborations may be leveraged for success.

## 2. Materials and Methods

### 2.1. The Monterey Bay Aquarium

The Monterey Bay Aquarium's mission is to inspire conservation of the ocean. Since opening its doors in 1984, MBA has advanced conservation through a fleet of animal husbandry, communications, education, exhibition, guest experience, marketing, policy, and research programs aimed at restoring and protecting California's ocean and coastal ecosystems. MBA has accomplished some of this work through rehabilitation efforts that advance the conservation of three imperiled California wildlife populations: the green sea turtle (*Chelonia mydas*), the Western snowy plover (*Charadrius nivosus nivosus*), and the southern sea otter. Although MBA engages in a variety of activities to support its conservation goals, in this paper we focus on its contributions to southern sea otter recovery through its Sea Otter Program.

### 2.2. Methods

The MBA Sea Otter Program (hereafter, 'program') consists of multiple components including stranding response, rehabilitation, population monitoring, research, husbandry, and veterinary care. To achieve the program's overall goal of southern sea otter recovery during the last 38 years, these activities have been conducted in collaboration with state and federal agencies, other non-profits, and universities. Although all of the program's efforts are equally important, this review focuses on the history of sea otter rehabilitation at

MBA with an emphasis on aspects that may inform other wildlife rehabilitation programs. We do so by highlighting successes and failures at critical stages that were formative in the program's evolution (Figure 2), summarizing southern sea otter stranding data, and providing several examples of how collaboration has informed advancement in scientific research.

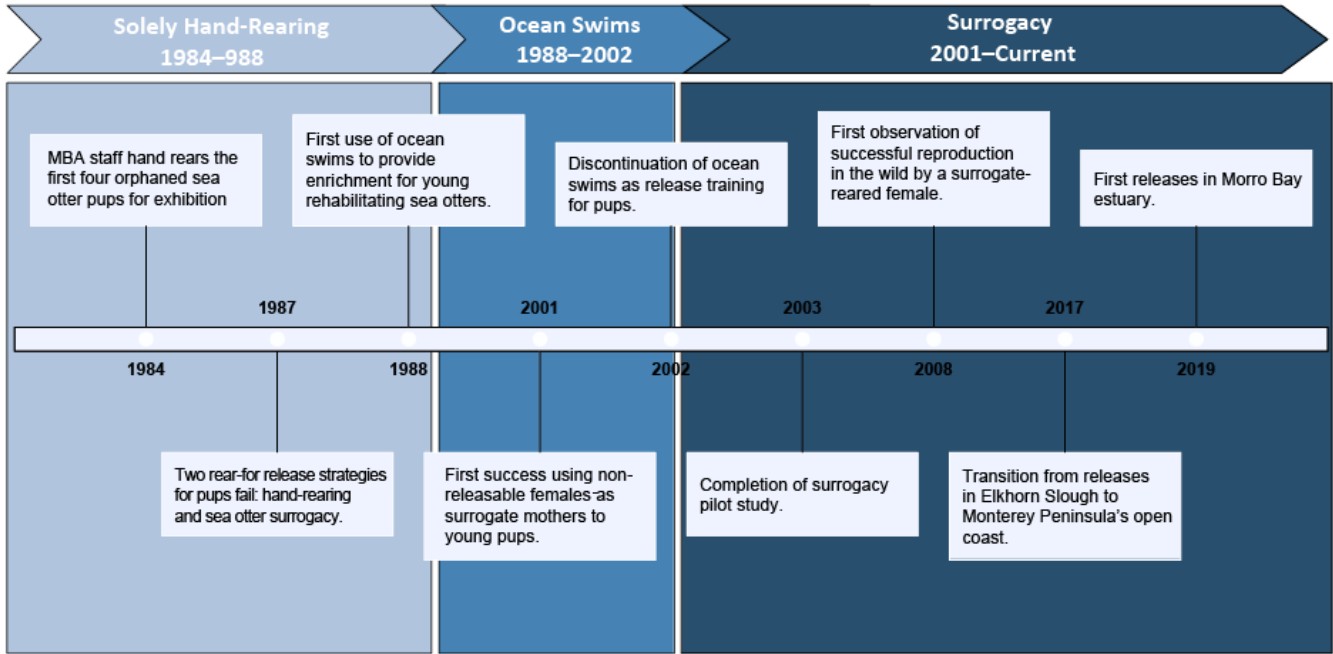

**Figure 2.** A timeline of key successes and failures through the history of MBA's Sea Otter Program rehabilitation efforts, marked by transitions in methods (i.e., hand-rearing, ocean swims, surrogacy).

## 3. Results

### 3.1. The Program's Beginnings

MBA rescued its first sea otters, when four pups separately stranded along the central California coast between February and April of 1984, half a year before the aquarium opened to the public (Figure 2). Aquarium staff hand-reared these pups, and because of this intensive care by humans, the USFWS deemed the pups non-releasable and authorized MBA to provide long-term care while exhibiting the otters. After acquiring these initial animals, the program continued to rescue, care for, and when possible, release stranded sea otters of all ages. While it was not the original plan to rehabilitate or exhibit sea otters in this new aquarium, support for the local sea otter population and the public's desire to help stranded individuals was clear.

As the number of sea otters in need of rescue continued to increase (Figure 3), program strategies had to evolve. Although animals of all age classes stranded, the majority of rescued older sea otters were in very poor health condition and only 22% (*n* = 146) were able to be rehabilitated from a wide variety of stranding causes (Table 1).

By contrast, pups stranded in relatively good health and responded well to treatment. With limited options for long-term placement of orphaned pups at other zoos and aquariums, and a wild population still well below recovery, program staff had to explore strategies for rearing pups for release to the wild. At the time, sea otter rearing strategies were unknown, but staff explored two different rear-for-release methods in 1987 (Figure 2). One case involved hand-rearing a pup prior to release, and in the other, an attempt was made to establish a bond between a pup and a resident non-releasable adult female sea otter. Neither strategy was successful, and program staff decided to continue to provide human care but hypothesized that exposing pups to their natural environment during development could aid in achieving more successful release outcomes. To do so, staff established strong bonds with individual animals and conducted frequent ocean enrich-

ment swims, which allowed the pups to explore nearshore rocky reefs and subtidal kelp forests in Monterey Bay. The intent of these swims was to encourage prey identification, foraging, and socialization with other wild otters in preparation for release as juveniles. Although ocean swims provided enrichment and allowed young animals to develop diving, swimming, and foraging skills, the close connection with human caregivers prevented many of the pups from establishing a natural wariness of people and reintegrating with the wild population after release [25].

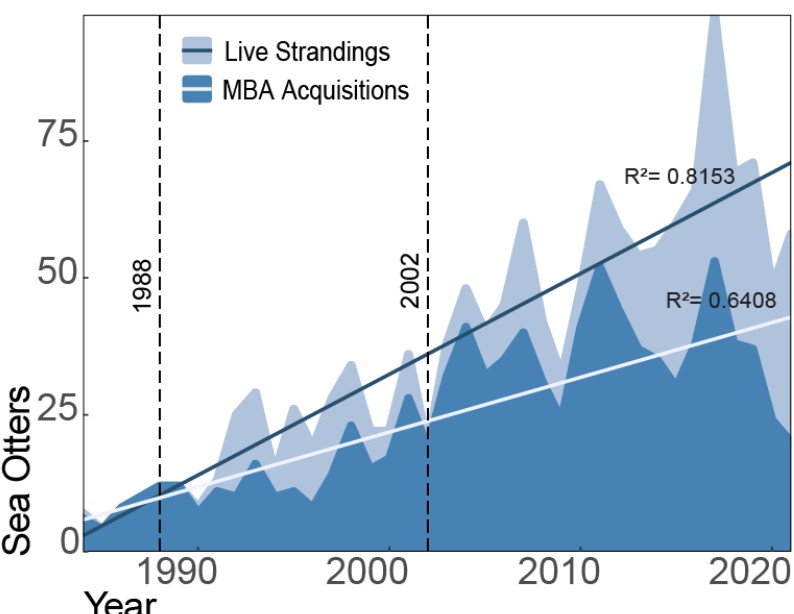

**Figure 3.** The number of sea otters that strand alive has increased over time, with annual fluctuations ($R^2$ = 0.8153). Likewise, the number of strandings responded to by the program has also increased over it's history ($R^2$ = 0.6408). From the start of the program, the caseload has increased from five animals in 1984 to nineteen at the start of surrogacy methods in 2002. From 1988 to 2002, when the program focused on hand-rearing and conducting enrichment ocean swims with sea otter pups, staff managed a caseload total of 213 animals, averaging 15 sea otters a year.

**Table 1.** Stranding causes of adult, subadult, and juvenile sea otters rescued, rehabilitated, and released throughout the program's history (*n* = 146, see also Nicholson et al. 2018).

| Stranding Cause | Description | No. Cases |
|---|---|---|
| Pre-mature weaning | Signs of emaciation or stunting due to early weaning or chronic malnutrition during weaning | 38 |
| ELS/mating trauma | Significant abrading or loss of nose-pad due to mating trauma (28), emaciation related to late-lactation (3) | 31 |
| Anthropogenic Trauma | Laceration, punctures, and fractures consistent with boat strike (2), fisheries interactions/line entanglement (6), tar or oiling (2), nonspecific anthropogenic trauma (5) | 15 |
| Neurological disease | Fine muscle tremors, seizures, and/or loss of motor function | 14 |
| Acanthocephalan peritonitis | Presence of acanthocephalan parasites | 11 |
| Shark bite | Lacerations from attempted depredation | 11 |
| Geriatric | Pathology related to old age and poor dentition | 4 |
| Other | Storm stress (1), respiratory disease (2), reproductive complications (1) or unknown (18) | 22 |
| Total | | 146 |

To closely monitor outcomes in the wild, sea otters receive intraperitoneal radio transmitter implants [27], and staff track them intensively over a two-week post-release period. Overall, sea otters released to the wild survived at rates comparable to their wild counterparts [25], except for young pups. By comparison, only 27% of individuals stranding as pups successfully reacclimated to the wild, while most required recapture and permanent placement in zoological facilities because of a failure to forage successfully or maintain a wariness of people [25,28].

As program staff explored how to successfully return young otters to the wild, its caseload of all ages continued to increase from six animals in 1984 to nineteen in 2002 (Figure 3). Despite the labor-intensive nature of veterinary treatment, care, and enrichment swims, the program admitted 213 live-stranded sea otters during this period (1988–2002), rehabilitating and releasing just under 50%. Through ongoing studies of the wild population, post-release tracking, and accumulation of data quantifying survival and reproduction rates of released individuals of all ages, program staff identified the need for a shift in rehabilitation methods to increase successful outcomes of stranded pups. Additionally, this decision process was aided by findings from a Blue Ribbon Panel [29] (a panel of subject matter experts) in 2002, which provided a framework for caseload management and clear integration of research and conservation priorities to best utilize limited resources.

### 3.2. Surrogacy

After an early failure while pairing a pup with an adult female sea otter, starting in 2001 the program re-explored using non-releasable females as surrogate mothers to rehabilitate dependent pups. This strategy was reimplemented to limit interactions with humans and leverage the natural maternal instincts of adult females to provide species-specific care to sea otter pups. Based on historical cases of sea otter adoption [30,31], staff believed this method would allow pups to better integrate with the wild population and avoid humans following release. The first successful introduction between an orphaned pup and a wild female in long-term care at MBA occurred in 2001 (Figure 2). Since then, the program has used surrogacy as a method to rear stranded pups for release.

Sea otter surrogacy involves five key stages; (1) stranding response, (2) stabilization, (3) surrogate rearing, (4) release preparation, and (5) post-release monitoring [32]. During stabilization, staff provide hands-on care while wearing disguises to minimize pup habituation with humans. At this stage, pups develop basic grooming, diving, and foraging skills before introductions to a surrogate female sea otter at approximately 8–10 weeks of age [16]. Once introductions are successful, mother and pup remain together during dependency with limited human intervention [16]. At around 6 months of age, the pup is weaned (i.e., permanently separated) from its surrogate, and veterinary staff administer several health exams in preparation for release [16]. Along with a VHF radio transmitter, released otters are instrumented with a unique color and placement combination of hind-flipper tags for identification in the field [32,33]. To assess how individuals are adjusting to the wild, post-release monitoring in collaboration with TMMC, CDFW, and USGS, details an otter's daily location, distance traveled, foraging success, behavior, and body condition.

With the development of these protocols, the focus of the program's pup-rearing efforts shifted from human-facilitated enrichment in the natural environment to surrogate-fostering of natural behaviors such as diving, grooming, foraging, and socializing. To assess whether this new method would result in greater success for released individuals compared with previous hand-rearing methods, the program conducted a trial study of five surrogate-reared pups that stranded from 2001 to 2003 [25] and were subsequently released into Elkhorn Slough. This release site is a seven-mile-long estuary located approximately 20 miles north of MBA along the center of the Monterey Bay coastline as shown in Figure 1.

During the study, surrogate-reared pups integrated well into the wild population, foraged successfully, and maintained their wariness of people [25]. The overall survival rate one-year post release was 71%, which was significantly higher than the rate of success (27%) prior to surrogacy [25]. These findings supported the shift in the program's approach

to rearing stranded pups and demonstrated the importance of post-release monitoring and the scientific method for informing program development. Over the course of the program's history, MBA has responded to 80% of all live stranded pups (Figure 4), and in recent years, as the numbers of live stranded sea otters of all age classes has increased, collaborators such as TMMC have been rehabilitating more mature animals, allowing MBA to focus on increasing its capacity for rehabilitating pups.

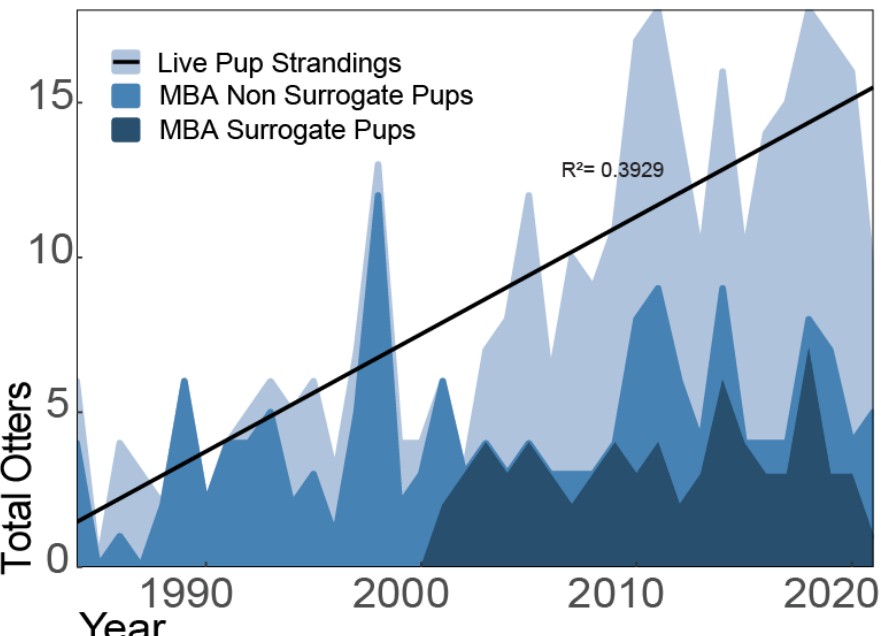

**Figure 4.** Throughout the program's history, live pup strandings have steadily increased ($R^2$ = 0.3929), and MBA responded to approximately 80% of these events. 50% of orphaned pups received extended care at the aquarium prior to release or transfer to a long-term care facility. From its implementation in 2002, only 28% of qualified pups (aged <10 weeks at stranding) were surrogate-reared. Criteria for aging pups at this young stage included size (total length), weight, tooth eruption patterns, and presence of natal pelage. Based on these methods, pup aging error is estimated at +/− 2 weeks.

### 3.3. Conservation Impact of Releasing Surrogate-Reared Sea Otters

Through rigorous data collection, a series of collaborative scientific research projects documented the positive conservation impacts of this program on local populations and ecosystems. By conducting long-term monitoring of released sea otters (n = 37) from 2002 to 2015 as well as annual monitoring of the population of sea otters in Elkhorn Slough, researchers were able to model the effect of this rehabilitation program on population growth [16]. In 2002, the population in Elkhorn Slough was approximately 20 individuals, and by 2015 it had grown to nearly 150 [34]. Based on release numbers and demographics, as well as observed and estimated survival and reproductive rates of rehabilitated and wild sea otters, surrogate-reared animals and their offspring were estimated to account for 55% of overall population growth in Elkhorn Slough [16]. Additionally, the release of female sea otters to this area likely contributed to a shift in the population from an area dominated by transient males, to a stable reproductive population. In much the same way that the presence of sea otters helps maintain kelp forests, increasing numbers of sea otters in Elkhorn Slough supported eelgrass (*Zostera* spp.) growth and recovery from eutrophic conditions, aiding in ecosystem restoration for other species in this habitat [17]. Specifically researchers identified a trophic cascade involving sea otters, crabs, and sea slugs that enhanced grazing of epiphytic algae from eelgrass blades, improving health and density of seagrass meadows [17]. Compared to measurements prior to the recolonization of sea otters, eelgrass extent increased 60% [35]. These outcomes have provided scientific support

for surrogacy as a rehabilitation method and for releasing these rehabilitated individuals to enhance population recovery and ecosystem restoration.

### 3.4. Collaborative Conservation Research

Beyond the direct population and ecosystem benefits of releasing sea otters, rehabilitation has also contributed to conservation efforts by informing research and development of veterinary and animal welfare protocols. This includes clinical diagnosis and treatment [36–39], adaptation of methods from techniques and technologies of other aspects of medicine to sea otters [40–44], expansion of the idiosyncrasies of pharmacology [45], and development/maintenance of a robust archive of biological materials sampled from sea otters over several decades [46–49].

Rehabilitation of sea otters at MBA has also inspired research to advance sea otter husbandry [50], oiled animal care [39,51], and knowledge of sea otter biology and physiology [23,41,52–55]. Findings from these collaborative research projects may inform wildlife managers about how best to anticipate, overcome, and prevent, risks to sea otters throughout a range of scenarios within wild and long-term care settings.

Sea otters undergoing rehabilitation have participated in collaborative experimental research that would be impossible to conduct in the wild. For example, during controlled exposure within monitored pools, sea otters demonstrated their vulnerability to drowning within commercial finfish and shellfish traps, which was a suspected population threat. This result led to a successful policy change that required the modification of fishing equipment with the addition of a rigid 5-inch ring to fyke openings, excluding most otters from entering and likely reducing sea otter entrapment [56]. These research advancements have either informed or directly aided sea otter conservation efforts and show how collaboration can magnify the benefits of wildlife rehabilitation.

## 4. Conclusions

### 4.1. Key Takeaway

MBA's Sea Otter Program exemplifies lessons learned and knowledge attained through rigorous research, methodological adaptation, and collaboration. Understanding how these factors have shaped the evolution of MBA's program can be helpful to other rehabilitation and conservation partnerships with similar goals [28]. The dissemination of information among programs can aid in addressing challenges in wildlife rehabilitation. The resource-intensive nature of wildlife rehabilitation is always a major obstacle, but collaboration provides alternative solutions and extends resources to achieve program goals. For MBA and other programs, increasing collaboration and partnerships continue to advance methods of rehabilitation, post-release monitoring, and analysis of impacts to wild populations, informing and improving conservation efforts into the future.

### 4.2. The Future

The evolution of this program resulted from a need to address the increase in live sea otter strandings, as well as the growing knowledge that rehabilitation methods should be informed by a scientific approach focused on improving individual animal and population welfare, and carried out in partnership with other groups and organizations. As the program continues to evolve, this knowledge and approach will be important when addressing future challenges in sea otter conservation.

Since the study at Elkhorn Slough, MBA's rehabilitation program and its partners have continued to successfully release animals in estuaries and open coast kelp forests within the southern sea otter's current range. Long-term survival of and reproduction by released sea otters have contributed to their species' keystone role of promoting healthier seagrass and kelp forest ecosystems [17,18,20]. These ecosystem benefits indicate that expanding releases for surrogate-reared sea otters could have local population-level and ecosystem benefits in other areas throughout the California coastline. Continued post-release monitoring of wild

sea otters is essential to understand how releasing otters within historical areas could aid population recovery and nearshore ecosystem restoration.

Although sea otter population abundance has increased along the central coast of California, the range extent of the population has remained mostly unchanged for the last decade (Figure 5). The growth of the existing wild population is limited by the lack of range expansion likely caused by shark bite mortality at the northern and southern peripheries [13]. Because range expansion may be critical for achieving population growth, MBA and its partners must focus their rehabilitation and release efforts in areas where population numbers are low to contribute to southern sea otter recovery.

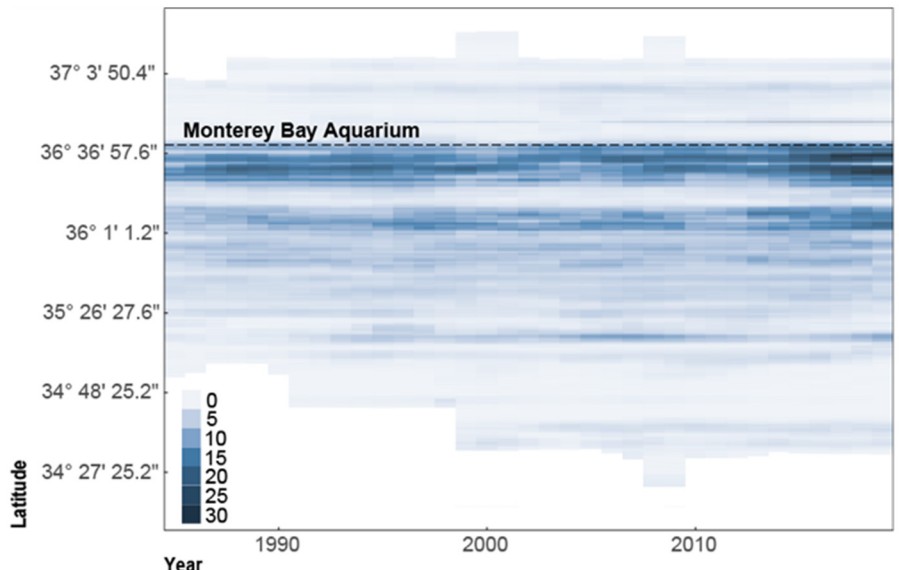

**Figure 5.** Southern sea otter population density; measured as otters per square kilometer across the extent of the subspecies range from 1985–2019. MBA is located at 36°37′4.8″ latitude [34].

Adult sea otters exhibit strong site fidelity [57], which creates challenges when attempting to rehabilitate and release them outside of the current range. MBA's method of releasing surrogate-reared juveniles could be a useful approach in addressing the population's lack of range expansion. Because an estimated 42% of releasable live strandings are young pups without an established home range, this age class presents an impactful rehabilitation opportunity. Since the start of the surrogacy program in 2002, MBA has only reared 28% of live stranded pups for release (Figure 4). Further partnerships with other zoos and aquariums could increase the capacity for surrogate-rearing orphaned pups, which may aid southern sea otter range expansion efforts. As the program continues to develop, existing and emerging partnerships will be vital in understanding if surrogacy could be applied to sea otter reintroduction efforts to advance sea otter recovery under the ESA.

**Author Contributions:** Conceptualization, J.A.F., L.K. and T.E.N.; Methodology, J.A.F., L.K. and C.Y.; Data Curation, L.K., T.E.N. and C.Y.; Writing—Original Draft Preparation, J.A.F., L.K. and T.E.N.; Writing—Review & Editing, J.A.F., S.H., A.B.J., L.K., K.A.M., M.J.M., T.E.N., M.M.S. and C.Y.; Visualization, L.K.; Supervision, J.A.F. and T.E.N. All authors have read and agreed to the published version of the manuscript.

**Funding:** This research was supported by generous donor contributions to the Monterey Bay Aquarium.

**Institutional Review Board Statement:** This study operated under U.S. Fish & Wildlife Service permits #MA032027-2 and LOA032027-2, and Monterey Bay Aquarium permits #MA186914 and #032027. All animal handling was in accordance with the requirements of USDA Class R Research Facility license # 93-R-0476.

**Data Availability Statement:** Not applicable.

**Acknowledgments:** Jack Ames, Greta Austin, Gena Bentall, Allie Bondi-Taylor, Sue Campbell, Dave Casper, Jennifer Coffey, Chris DeAngelo, James Estes, Katie Finch, Krista Hanni, Mike Harris, Katie Hawkins, Julie Hymer, Michele Jefferies, Sonny Knaub, Erin Lenihan, Kaitlyn Reilly, Marianne Riedman, Julie Stewart, Kattie Stong, Candace Tahara, Marcie Tarvid, Bob VanWagenen, Thomas Williams, Linda Yingling, Marissa Young, and the many staff, volunteers, and interns past and present at MBA and TMMC who helped care for, and track the many animals in the program.

**Conflicts of Interest:** The authors declare no conflict of interest.

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
