# Peer review of "Southern Sea Otter Rehabilitation: Lessons and Impacts from the Monterey Bay Aquarium"

_2673-5636, doi:10.3390/jzbg3040047_

Round 1

Reviewer 1 Report

The work is written in a manner consistent with a separate historical introduction, which gives the basis for understanding and appreciating the content.

The need to create, maintain and, above all, develop such centers to help wild animals is clearly described. The need to renew natural resources by restoring valuable otters to the wild was indicated.

However, the work lacked a list of rehabilitation programs as such. It is obvious to me that each time it required individual recognition of the animal's needs, but the quantitative and qualitative combination would definitely enrich the manuscript.

A tabular list or through graphs could describe the number of animals in the years that underwent rehabilitation, of which young otters which required nutrition and pediatric care, adult individuals perhaps divided into sex with a description of qualitative needs such as dental diseases, injuries, skin diseases, amputations, etc.. Such an approach would definitely enrich the content and give a detailed insight into the activities carried out.

The beginning of such a sample is illustrated in Figures 3 and 4 and sufficiently described. It seems, however, that a clear statistical emphasis on the (apparent) trend will strengthen the manuscript scientifically.

If the authors do not have qualitative data, then for a quantitative approach expressed by the trends of phenomena (in this case, the number of animals helped, the number of animals restored to life in the natural environment) in terms of the trend in years, with the indication of the R-factor and the trend equation, it seems necessary.

After supplementing the content with a possible statistical approach, the manuscript deserves a positive assessment and attention of the readers.

Reviewer 2 Report

This is a well written and well-organized description of a conservation program. There are opportunities for this be more informative about certain aspects of rehab and release of this species (the weaning process, soft/hard release, etc…) however that may be beyond the scope of this special edition. I think this manuscript is acceptable in its current form.
